# You CAN Teach an Old Dog New Tricks!
# On Training Knowledge Graph Embeddings

**Daniel Ruffinelli**\*
Data and Web Science Group
University of Mannheim, Germany

**Samuel Broscheit**\*
Data and Web Science Group
University of Mannheim, Germany

**Rainer Gemulla**†
Data and Web Science Group
University of Mannheim, Germany

## Abstract

Knowledge graph embedding (KGE) models learn algebraic representations of the entities and relations in a knowledge graph. A vast number of KGE techniques for multi-relational link prediction have been proposed in the recent literature, often with state-of-the-art performance. These approaches differ along a number of dimensions, including different model architectures, different training strategies, and different approaches to hyperparameter optimization. In this paper, we take a step back and aim to summarize and quantify empirically the impact of each of these dimensions on model performance. We report on the results of an extensive experimental study with popular model architectures and training strategies across a wide range of hyperparameter settings. We found that when trained appropriately, the relative performance differences between various model architectures often shrinks and sometimes even reverses when compared to prior results. For example, RESCAL (Nickel et al., 2011), one of the first KGE models, showed strong performance when trained with state-of-the-art techniques; it was competitive to or outperformed more recent architectures. We also found that good (and often superior to prior studies) model configurations can be found by exploring relatively few random samples from a large hyperparameter space. Our results suggest that many of the more advanced architectures and techniques proposed in the literature should be revisited to reassess their individual benefits. To foster further reproducible research, we provide all our implementations and experimental results as part of the open source LibKGE framework.

## 1 Introduction

Knowledge graph embedding (KGE) models learn algebraic representations, termed embeddings, of the entities and relations in a knowledge graph. They have been successfully applied to knowledge graph completion (Nickel et al., 2015) as well as in downstream tasks and applications such as recommender systems (Wang et al., 2017) or visual relationship detection (Baier et al., 2017).

A vast number of different KGE models for multi-relational link prediction have been proposed in the recent literature; e.g., RESCAL (Nickel et al., 2011), TransE (Bordes et al., 2013), DistMult, ComplEx (Trouillon et al., 2016), ConvE (Dettmers et al., 2018), TuckER (Balazevic et al., 2019), RotatE (Sun et al., 2019a), SACN (Shang et al., 2019), and many more. Model architectures generally differ in the way the entity and relation embeddings are combined to model the presence or absence of an edge (more precisely, a subject-predicate-object triple) in the knowledge graph; they include factorization models (e.g., RESCAL, DistMult, ComplEx, TuckER), translational models (TransE, RotatE), and more advanced models such as convolutional models (ConvE). In many cases, the introduction of new models went along with new approaches for training these models—e.g., new training types (such as negative sampling or 1vsAll scoring), new loss functions (such as

---

\*Contributed equally. `{daniel,broscheit}@informatik.uni-mannheim.de`
†`rgemulla@uni-mannheim.de`

| Publication | Model | Loss | Training | Regularizer | Optimizer | Reciprocal |
|---|---|---|---|---|---|---|
| Nickel et al. (2011) | **RESCAL** | MSE | Full | **L2** | ALS | No |
| Bordes et al. (2013) | **TransE** | **MR** | **NegSamp** | **Normalization** | **SGD** | No |
| Yang et al. (2015) | **DistMult** | MR | NegSamp | **Weighted** L2 | **Adagrad** | No |
| Trouillon et al. (2016) | **ComplEx** | **BCE** | NegSamp | Weighted L2 | Adagrad | No |
| Kadlec et al. (2017) | DistMult | **CE** | NegSamp | Weighted L2 | **Adam** | No |
| Dettmers et al. (2018) | **ConvE** | BCE | **KvsAll** | **Dropout** | Adam | **Yes** |
| Lacroix et al. (2018) | ComplEx | CE | **1vsAll** | Weighted **L3** | Adagrad | Yes |

MSE = mean squared error, MR = margin ranking, BCE = binary cross entropy, CE = cross entropy

Table 1: Selected KGE models and training strategies from the literature. Entries marked in bold were introduced (or first used) in the context of KGE in the corresponding publication.

pairwise margin ranking or binary cross entropy), new forms of regularization (such as unweighted and weighted L2), or the use of reciprocal relations (Kazemi & Poole, 2018; Lacroix et al., 2018)— and ablation studies were not always performed. Table 1 shows an overview of selected models and training techniques along with the publications that introduced them.

The diversity in model training makes it difficult to compare performance results for various model architectures, especially when results are reproduced from prior studies that used a different experimental setup. Model hyperparameters are commonly tuned using grid search on a small grid involving hand-crafted parameter ranges or settings known to "work well" from prior studies. A grid suitable for one model may be suboptimal for another, however. Indeed, it has been observed that newer training strategies can considerably improve model performance (Kadlec et al., 2017; Lacroix et al., 2018; Salehi et al., 2018).

In this paper, we take a step back and aim to summarize and quantify empirically the impact of different model architectures and different training strategies on model performance. We performed an extensive set of experiments using popular model architectures and training strategies in a common experimental setup. In contrast to most prior work, we considered many training strategies as well as a large hyperparameter space, and we performed model selection using quasi-random search (instead of grid search) followed by Bayesian optimization. We found that this approach was able to find good (and often superior to prior studies) model configurations with relatively low effort. Our study complements and expands on the results of Kotnis & Nastase (2018) (focus on negative sampling) and Mohamed et al. (2019) (focus on loss functions) as well as similar studies in other areas, including language modeling (Melis et al., 2017), generative adversarial networks (Lucic et al., 2018), or sequence tagging (Reimers & Gurevych, 2017).

We found that when trained appropriately, the performance of a particular model architecture can by far exceed the performance observed in older studies. For example, RESCAL (Nickel et al., 2011), which constitutes one of the first KGE models but is rarely considered in newer work, showed very strong performance in our study: it was competitive to or outperformed more recent architectures such as ConvE (Dettmers et al., 2018) and TuckER (Balazevic et al., 2019). More generally, we found that the relative performance differences between various model architectures often shrunk and sometimes even reversed when compared to prior results. This suggests that (at least currently) training strategies have a significant impact on model performance and may account for a substantial fraction of the progress made in recent years. We also found that suitable training strategies and hyperparameter settings vary significantly across models and datasets, indicating that a small search grid may bias results on model performance. Fortunately, as indicated above, large hyperparameter spaces can be (and should be) used with little additional training effort. To facilitate such efforts, we provide implementations of relevant training strategies, models, and evaluation methods as part of the open source LibKGE framework,[1] which emphasizes reproducibility and extensibility.

Our study focuses solely on pure KGE models, which do not exploit auxiliary information such as textual data or logical rules (Wang et al., 2017). Since many of the studies on these non-pure models did not (and, to be fair, could not) use current training strategies and consequently underestimated the performance of pure KGE models, their results and conclusions need to be revisited.

---

[1] https://github.com/uma-pi1/kge

## 2 KNOWLEDGE GRAPH EMBEDDINGS: MODELS, TRAINING, EVALUATION

The literature on KGE models is expanding rapidly. We review selected architectures, training methods, and evaluation protocols; see Table 1. The table examplarily indicates that new model architectures are sometimes introduced along with new training strategies (marked bold). Reasonably recent survey articles about KGE models include Nickel et al. (2015) and Wang et al. (2017).

**Multi-relational link prediction.** KGE models are typically trained and evaluated in the context of multi-relational link prediction for knowledge graphs (KG). Generally, given a set $\mathcal{E}$ of entities and a set $\mathcal{R}$ of relations, a knowledge graph $\mathcal{K} \subseteq \mathcal{E} \times \mathcal{R} \times \mathcal{E}$ is a set of subject-predicate-object (spo) triples. The goal of multi-relational link prediction is to "complete the KG", i.e., to predict true but unobserved triples based on the information in $\mathcal{K}$. Common approaches include rule-based methods (Galarraga et al., 2013; Meilicke et al., 2019), KGE methods (Nickel et al., 2011; Bordes et al., 2013; Trouillon et al., 2016; Dettmers et al., 2018), and hybrid methods (Guo et al., 2018).

**Knowledge graph embeddings (KGE).** A KGE model associates with each entity $i \in \mathcal{E}$ and relation $k \in \mathcal{R}$ an *embedding* $\boldsymbol{e}_i \in \mathbb{R}^{d_e}$ and $\boldsymbol{r}_k \in \mathbb{R}^{d_r}$ in a low-dimensional vector space, respectively; here $d_e, d_r \in \mathbb{N}^+$ are hyperparameters for the *embedding size*. Each particular model uses a *scoring function* $s : \mathcal{E} \times \mathcal{R} \times \mathcal{E} \rightarrow \mathbb{R}$ to associate a *score* $s(i, k, j)$ with each potential triple $(i, k, j) \in \mathcal{E} \times \mathcal{R} \times \mathcal{E}$. The higher the score of a triple, the more likely it is considered to be true by the model. The scoring function takes form $s(i, k, j) = f(\boldsymbol{e}_i, \boldsymbol{r}_k, \boldsymbol{e}_j)$, i.e., depends on $i$, $k$, and $j$ only through their respective embeddings. Here $f$, which represents the model architecture, may be either a fixed function or learned (e.g., $f$ may be a parameterized function).

**Evaluation.** The most common evaluation task for KGE methods is *entity ranking*, which is a form of question answering. The available data is partitioned into a set of training, validation, and test triples. Given a test triple $(i, k, j)$ (unseen during training), the entity ranking task is to determine the correct answer—i.e., the missing entity $j$ and $i$, resp.—to questions $(i, k, ?)$ and $(?, k, j)$. To do so, potential answer triples are first ranked by their score in descending order. All triples but $(i, k, j)$ that occur in the training, validation, or test data are subsequently filtered out so that other triples known to be true do not affect the ranking. Finally, metrics such as the mean reciprocal rank (MRR) of the true answer or the average HITS@$k$ are computed; see the appendix.

**KGE models.** KGE model architectures differ in their scoring function. We can roughly classify models as *decomposable* or *monolithic*: the former only allow element-wise interactions between (relation-specific) subject and object embeddings, whereas the latter allow arbitrary interactions. More specifically, decomposable models use scoring functions of form $s(i, k, j) = f(\sum_z g([h_1(\boldsymbol{e}_i, \boldsymbol{e}_r) \circ h_2(\boldsymbol{e}_r, \boldsymbol{e}_j)]_z))$, where $\circ$ is any element-wise function (e.g., multiplication), $h_1$ and $h_2$ are functions that obtain *relation-specific* subject and object embeddings, resp., and $g$ and $f$ are scalar functions (e.g., identity or sigmoid). The most popular models in this category are perhaps RESCAL (Nickel et al., 2011), TransE (Bordes et al., 2013), DistMult (Yang et al., 2015), ComplEx (Trouillon et al., 2016), and ConvE (Dettmers et al., 2018). RESCAL's scoring function is bilinear in the entity embeddings: it uses $s(i, k, j) = \boldsymbol{e}_i^T \boldsymbol{R}_k \boldsymbol{e}_j$, where $\boldsymbol{R}_k \in \mathbb{R}^{d_e \times d_e}$ is a matrix formed from the entries of $\boldsymbol{r}_k \in \mathbb{R}^{d_r}$ (where $d_r = d_e^2$). DistMult and ComplEx can be seen as constrained variants of RESCAL with smaller relation embeddings ($d_r = d_e$). TransE is a translation-based model and uses negative distance $-\|\boldsymbol{e}_i + \boldsymbol{r}_k - \boldsymbol{e}_j\|_p$ between $\boldsymbol{e}_i + \boldsymbol{r}_k$ and $\boldsymbol{e}_j$ as score, commonly using the L1 norm ($p = 1$) or the L2 norm ($p = 2$). Finally, ConvE uses a 2D convolutional layer and a large fully connected layer to obtain relation-specific entity embeddings (i.e., in $h_1$ above). Other recent examples for decomposable models include TuckER (Balazevic et al., 2019), RotatE (Sun et al., 2019a), and SACN (Shang et al., 2019). Decomposable models are generally fast to use: once the relation-specific embeddings are (pre-)computed, score computations are cheap. Monolithic models—such as ConvKB or KBGAT—do not decompose into relation-specific embeddings: they take form $s(i, k, j) = f(\boldsymbol{e}_i, \boldsymbol{r}_k, \boldsymbol{e}_j)$. Such models are more flexible, but they are also considerably more costly to train and use. It is currently unclear whether monolithic models can achieve comparable or better performance than decomposable models Sun et al. (2019b).

**Training type.** There are three commonly used approaches to train KGE models, which differ mainly in the way negative examples are generated. First, training with negative sampling (NegSamp) (Bordes et al., 2013) obtains for each positive triple $t = (i, k, j)$ from the training data a set of (pseudo-)negative triples obtained by randomly perturbing the subject, relation, or object position in $t$ (and optionally verifying that the so-obtained triples do not exist in the KG). An

alternative approach (Lacroix et al., 2018), which we term *1vsAll*, is to omit sampling and take *all* triples that can be obtained by perturbing the subject and object positions as negative examples for $t$ (even if these tuples exist in the KG). 1vsAll is generally more expensive than NegSamp, but it is feasible (and even surprisingly fast in practice) if the number of entities is not excessively large. Finally, Dettmers et al. (2018) proposed a training type that we term *KvsAll*[2]: this approach (i) constructs batches from non-empty rows $(i, k, *)$ or $(*, k, j)$ instead of from individual triples, and (ii) labels all such triples as either positive (occurs in training data) or negative (otherwise).

**Loss functions**. Several loss functions for training KGEs have been introduced so far. RESCAL originally used squared error between the score of each triple and its label (positive or negative). TransE used pairwise margin ranking with hinge loss (MR), where each pair consists of a positive triple and one of its negative triples (only applicable to NegSamp and 1vsAll) and the margin $\eta$ is a hyperparameter. Trouillon et al. (2016) proposed to use binary cross entropy (BCE) loss: it applies a sigmoid to the score of each (positive or negative) triple and uses the cross entropy between the resulting probability and that triple's label as loss. BCE is suitable for multi-class and multi-label classification. Finally, Kadlec et al. (2017) used cross entropy (CE) between the model distribution (softmax distribution over scores) and the data distribution (labels of corresponding triples, normalized to sum to 1). CE is more suitable for multi-class classification (as in NegSamp and 1vsAll), but it has also been used in the multi-label setting (KvsAll). Mohamed et al. (2019) found that the choice of loss function can have a significant impact on model performance, and that the best choice is data and model dependent. Our experimental study provides additional evidence for this finding.

**Reciprocal relations.** Kazemi & Poole (2018) and Lacroix et al. (2018) introduced the technique of *reciprocal relations* into KGE training. Observe that during evaluation and also most training methods discussed above, the model is solely asked to score subjects (for questions of form $(?, k, j)$) or objects (for questions of form $(i, k, ?)$). The idea of reciprocal relations is to use separate scoring functions $s_{\text{sub}}$ and $s_{\text{obj}}$ for each of these tasks, resp. Both scoring functions share entity embeddings, but they do not share relation embeddings: each relation thus has two embeddings.[3] The use of reciprocal relations may decrease the computational cost (as in the case of ConvE), and it may also lead to better model performance Lacroix et al. (2018) (e.g., for relations in which one direction is easier to predict). On the downside, the use of reciprocal relations means that a model does not provide a single triple score $s(i, k, j)$ anymore; generally, $s_{\text{sub}}(i, k, j) \neq s_{\text{obj}}(i, k, j)$. Kazemi & Poole (2018) proposed to take the average of the two triple scores and explored the resulting models.

**Regularization.** The most popular form of regularization in the literature is L2 regularization on the embedding vectors, either unweighted or weighted by the frequency of the corresponding entity or relation (Yang et al., 2015). Lacroix et al. (2018) proposed to use L3 regularization. TransE normalized the embeddings to unit norm after each update. ConvE used dropout (Srivastava et al., 2014) in its hidden layers (and only in those). In our study, we additionally consider L1 regularization and the use of dropout on the entity and/or relation embeddings.

**Hyperparameters.** Many more hyperparameters have been used in prior work. This includes, for example, different methods to initialize model parameters, different optimizers, different optimizer parameters such as the learning rate or batch size, the number of negative examples for NegSamp, the regularization weights for entities and relations, and so on. To deal with such a large search space, most prior studies favor grid search over a small grid where most of these hyperparameters remain fixed. As discussed before, this approach may lead to bias in the results, however.

## 3 EXPERIMENTAL STUDY

In this section, we report on the design and results of our experimental study. We focus on the most salient points here; more details can be found in the appendix.

---

[2]Note that the KvsAll strategy is called 1-N scoring in Dettmers et al. (2018).

[3]Alternatively, we can view such an approach as only predicting objects, but doing so for both each original relation ($k$) and a new reciprocal relation formed by switching subject and object ($k^{-1}$).

## 3.1 EXPERIMENTAL SETUP

**Datasets.** We used the FB15K-237 (Toutanova & Chen, 2015) (extracted from Freebase) and WNRR (Dettmers et al., 2018) (extracted from WordNet) datasets in our study. We chose these datasets because (i) they are frequently used in prior studies, (ii) they are "hard" datasets that have been designed specifically to evaluate multi-relational link prediction techniques, (iii) they are diverse in that relative model performance often differs, and (iv) they are of reasonable size for a large study. Dataset statistics are given in Table 4 in the appendix.

**Models.** We selected RESCAL (Nickel et al., 2011), TransE (Bordes et al., 2013), DistMult (Yang et al., 2015), ComplEx (Trouillon et al., 2016) and ConvE (Dettmers et al., 2018) for our study. These models are perhaps the most well-known KGE models and include both early and more recent models. We did not consider monolithic models due to their excessive training cost.

**Evaluation.** We report filtered MRR (%) and filtered HITS@10 (%); see Sec. 2 and the appendix for details. We use test data to compare final model performance (Table 2) and validation data for our more detailed analysis.

**Hyperparameters.** We used a large hyperparameter space to ensure sure that suitable hyperparameters for each model are not excluded a priori. We included all major training types (NegSamp, 1vsAll, KvsAll), use of reciprocal relations, loss functions (MR, BCE, CE), regularization techniques (none/L1/L2/L3, dropout), optimizers (Adam, Adagrad), and initialization methods (4 in total) used in the KGE community as hyperparameters. We considered three embeddings sizes (128, 256, 512) and used separate weights for dropout/regularization for entity and relation embeddings. Table 5 in the appendix provides additional details. To the best of our knowledge, no prior study used such a large hyperparameter search space.

**Training.** All models were trained for a maximum of 400 epochs. We validated models using filtered MRR (on validation data) every five epochs and performed early stopping with a patience of 50 epochs. To keep search tractable, we stopped training on models that did not reach $\geq 5\%$ filtered MRR after 50 epochs; in our experience, such configurations did not produce good models.

**Model selection.** Model selection was performed using filtered MRR on validation data. We used the Ax framework (`https://ax.dev/`) to conduct quasi-random hyperparameter search via a Sobol sequence. Quasi-random search methods aim to distribute hyperparameter settings evenly and try avoid "clumping" effects (Bergstra & Bengio, 2012). More specifically, for each dataset and model, we generated 30 different configurations per valid combination of training type and loss function (2 for TransE, which only supports NegSamp+MR and NegSamp+CE; 7 for all other models). After the quasi-random hyperparameter search, we added a short Bayesian optimization phase (best configuration so far + 20 new trials, using expected improvement; also provided by Ax) to tune the numerical hyperparameters further. Finally, we trained five models with the so-obtained hyperparameter configuration and selected the best-performing model according to validation MRR as the final model. The standard deviation of the validation MRR was relatively low; see Table 9.

**Reproducibility.** We implemented all models, training strategies, evaluation, and hyperparameter search in the LibKGE framework. The framework emphasizes reproducibility and extensibility and is highly configurable. We provide all configurations used in this study as well as the detailed data of our experiments for further analysis.[4] We hope that these resources facilitate the evaluation of KGE models in a comparable environment.

## 3.2 COMPARISON OF MODEL PERFORMANCE

Table 2 shows the filtered MRR and filtered Hits@10 on test data of various models both from prior studies and the best models (according filtered validation MRR) found in our study.

**First reported performance vs. our observed performance.** We compared the first reported performance on FB15K-237 and WNRR ("First" block of Table 2) with the performance obtained in our study ("Ours" block). We found that the performance of a single model can vary wildly across studies. For example, ComplEx was first run on FB15K-237 by Dettmers et al. (2018), where it achieved a filtered MRR of 24.7%. This is a relatively low number by today's standards. In our

---

[4]`https://github.com/uma-pi1/kge-iclr20`

|  |  | FB15K-237 | | WNRR | |
|---|---|---|---|---|---|
|  |  | MRR | Hits@10 | MRR | Hits@10 |
| *First* | RESCAL (Wang et al., 2019) | 27.0 | 42.7 | 42.0 | 44.7 |
|  | TransE (Nguyen et al., 2018) | 29.4 | 46.5 | 22.6 | 50.1 |
|  | DistMult (Dettmers et al., 2018) | 24.1 | 41.9 | 43.0 | 49.0 |
|  | ComplEx (Dettmers et al., 2018) | 24.7 | 42.8 | 44.0 | 51.0 |
|  | ConvE (Dettmers et al., 2018) | 32.5 | 50.1 | 43.0 | 52.0 |
| *Ours* | RESCAL | **35.7** | **54.1** | 46.7 | 51.7 |
|  | TransE | 31.3 | 49.7 | 22.8 | 52.0 |
|  | DistMult | 34.3 | 53.1 | 45.2 | 53.1 |
|  | ComplEx | 34.8 | 53.6 | **47.5** | **54.7** |
|  | ConvE | 33.9 | 52.1 | 44.2 | 50.4 |
| *Recent* | TuckER (Balazevic et al., 2019) | **35.8** | **54.4** | 47.0 | 52.6 |
|  | RotatE (Sun et al., 2019a) | 33.8 | 53.3 | **47.6** | **57.1** |
|  | SACN (Shang et al., 2019) | 35.0 | 54.0 | 47.0 | 54.4 |
| *Large* | DistMult (Salehi et al., 2018) | 35.7 | 54.8 | 45.5 | 54.4 |
|  | ComplEx-N3 (Lacroix et al., 2018) | **37.0** | **56.0** | **49.0** | **58.0** |

Table 2: Model performance in prior studies and our study (as percentages, on test data). *First*: first reported performance on the respective datasets (oldest models first); *Ours*: performance in our study; *Recent*: best performance results obtained in prior studies of selected recent models; *Large*: best performance achieved in prior studies using very large models (not part of our search space). Bold numbers indicate best performance in group. References indicate where the performance number was reported.

study, ComplEx achieved a competitive MRR of 34.8%, which is a large improvement over the first reports. Studies that report the lower performance number of ComplEx (i.e., 24.7%) thus do not adequately capture its performance. Similar remarks hold for RESCAL and DistMult as well as (albeit to a smaller extent) ConvE and TransE. To the best of our knowledge, our results for RESCAL and ConvE constitute the best results for these models obtained so far.

**Relative model performance (our study).** Next, we compared the performance across the models models used in our study ("Ours" block). We found that the relative performance differences between model architectures often shrunk and sometimes reversed when compared to prior results ("First" block). For example, ConvE showed the best overall performance in prior studies, but is consistently outperformed by RESCAL, DistMult, and ComplEx in our study. As another example, RESCAL (Nickel et al., 2011), which constitutes one of the first KGE models but is rarely considered in newer work, showed strong performance and outperformed all models except ComplEx. This suggests that (at least currently) training strategies have a significant impact on model performance and may account for a large fraction of the progress made in recent years.

**Relative model performance (overall).** Table 2 additionally shows the best performance results obtained in prior studies of selected recent models ("Recent" block) and very large models with very good performance ("Large" block). We found that TuckER (Balazevic et al., 2019), RotatE (Sun et al., 2019a), and SACN (Shang et al., 2019) all achieve state-of-the-art performance, but the performance difference to the best prior models ("Ours" block) in terms of MRR is small or even vanishes. Even for HITS@10, which we did not consider for model selection, the advantage of more recent models is often small, if present. The models in the last block ("Large") show the best performance numbers we have seen in the literature. For DistMult and ComplEx, these numbers have been obtained using very large embedding sizes (up to 4000). Such large models were not part of our search space.

**Limitations.** We note that all models used in our study are likely to yield better performance when hyperparameter tuning is further improved. For example, we found a ComplEx configuration (of size 100) that achieved 49.0% MRR and 56.5% Hits@10 on WNRR, and a RESCAL configuration that achieved 36.3% MRR and 54.6% Hits@10 on FB15K-237 in preliminary experiments. Sun et al. (2019a) report on a TransE configuration that achieved 33.3% MRR (52.2% Hits@10) on

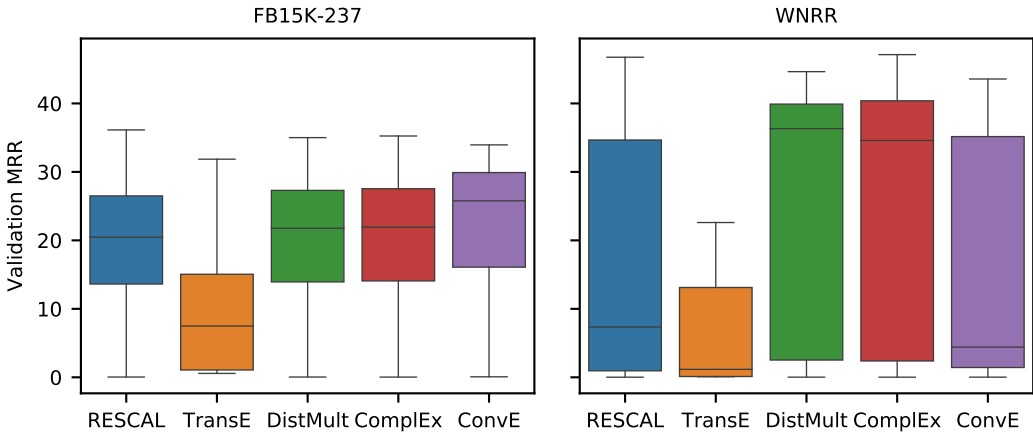

Figure 1: Distribution of filtered MRR (%) on validation data over the quasi-random hyperparameter configurations explored in our study.

FB15K-237. Since the focus of this study is to compare models in a common experimental setup and without manual intervention, we do not further report on these results.

### 3.3 IMPACT OF HYPERPARAMETERS

**Anatomy of search space.** Figure 1 shows the distribution of filtered MRR for each model and dataset (on validation data, Sobol configurations only). Each distribution consists of roughly 200 different quasi-random hyperparameter configurations (except TransE, for which 60 configurations have been used). Perhaps unsurprisingly, we found that all models showed a wide dispersion on both datasets and only very few configurations achieved state-of-the-art results. There are, however, notable differences across datasets and models. On FB15K-237, for example, the median ConvE configuration is best; on WNRR, DistMult and Complex have a much higher median MRR. Generally, the impact of the hyperparameter choice is more pronounced on WNRR (higher variance) than on FB15K-237.

**Best configurations (quasi-random search).** The hyperparameters of the best performing models during our quasi-random search are reported in Table 3 (selected hyperparameters) and Tables 6 and 7 (in the appendix, all hyperparameters). First, we found that the optimum choice of hyperparameters is often model and dataset dependent: with the exception of the loss function (discussed below), almost no hyperparameter setting was consistent across datasets and models. For example, the NegSample training type was frequently selected on FB15K-237 but not on WNRR, where KvsAll was more prominent. Our findings provide further evidence that grid search on a small search space is not suitable to compare model performance because the result may be heavily influenced by the specific grid points being used. Moreover, the budget that we used for quasi-random search was comparable to a grid search over a small (or at least not very large) grid: we tested merely 30 different model configurations for every combination of training type and loss function. It is therefore both feasible and beneficial to work with a large search space.

**Best configurations (Bayesian optimization).** We already obtained good configurations solely from quasi-random search. After Bayesian optimization (which we used to tune the numerical hyperparameters), the performance on test data almost always improved only marginally. The best hyperparameters after the Bayesian optimization phase are reported in Table 8 in the appendix.

**Impact of specific hyperparameters.** It is difficult to asses the impact of each hyperparameter individually. As a proxy for the importance of each hyperparameter setting, we report in Table 3 and Tables 6 and 7 (appendix) the best performance in terms of filtered MRR on validation data for a *different choice* of value for each hyperparameter (difference in parenthesis). For example, the best configuration during quasi-random search for RESCAL on FB15K-237 did not use reciprocal relations. The best configuration that did use reciprocal relations had an MRR that was 0.5 points lower (35.6 instead of 36.1). This does not mean that the gap is explained by the use of the reciprocal relations alone, as other hyperparameters may also be different, but it does show the best performance

|  |  | RESCAL | TransE | DistMult | ComplEx | ConvE |
|---|---|---|---|---|---|---|
| *FB15K-237* | Valid. MRR | *36.1* | *31.5* | *35.0* | *35.3* | *34.3* |
|  | Emb. size | 128 *(-0.5)* | 512 *(-3.7)* | 256 *(-0.2)* | 256 *(-0.3)* | 256 *(-0.4)* |
|  | Batch size | 512 *(-0.5)* | 128 *(-7.1)* | 1024 *(-0.2)* | 1024 *(-0.3)* | 1024 *(-0.4)* |
|  | Train type | 1vsAll *(-0.8)* | NegSamp – | NegSamp *(-0.2)* | NegSamp *(-0.3)* | 1vsAll *(-0.4)* |
|  | Loss | CE *(-0.9)* | CE *(-7.1)* | CE *(-3.1)* | CE *(-3.8)* | CE *(-0.4)* |
|  | Optimizer | Adam *(-0.5)* | Adagrad *(-3.7)* | Adagrad *(-0.2)* | Adagrad *(-0.5)* | Adagrad *(-1.5)* |
|  | Initializer | Normal *(-0.8)* | XvNorm *(-3.7)* | Unif. *(-0.2)* | Unif. *(-0.5)* | XvNorm *(-0.4)* |
|  | Regularizer | None *(-0.5)* | L2 *(-3.7)* | L3 *(-0.2)* | L3 *(-0.3)* | L3 *(-0.4)* |
|  | Reciprocal | No *(-0.5)* | Yes *(-9.5)* | Yes *(-0.3)* | Yes *(-0.3)* | Yes – |
| *WNRR* | Valid. MRR | *46.8* | *22.6* | *45.4* | *47.6* | *44.3* |
|  | Emb. size | 128 *(-1.0)* | 512 *(-5.1)* | 512 *(-1.1)* | 128 *(-1.0)* | 512 *(-1.2)* |
|  | Batch size | 128 *(-1.0)* | 128 *(-5.1)* | 1024 *(-1.1)* | 512 *(-1.0)* | 1024 *(-1.3)* |
|  | Train type | KvsAll *(-1.0)* | NegSamp – | KvsAll *(-1.1)* | 1vsAll *(-1.0)* | KvsAll *(-1.2)* |
|  | Loss | CE *(-2.0)* | CE *(-5.1)* | CE *(-2.4)* | CE *(-3.5)* | CE *(-1.4)* |
|  | Optimizer | Adam *(-1.2)* | Adagrad *(-5.8)* | Adagrad *(-1.5)* | Adagrad *(-1.5)* | Adam *(-1.4)* |
|  | Initializer | Unif. *(-1.0)* | XvNorm *(-5.1)* | Unif. *(-1.3)* | Unif. *(-1.5)* | XvNorm *(-1.4)* |
|  | Regularizer | L3 *(-1.2)* | L2 *(-5.1)* | L3 *(-1.1)* | L2 *(-1.0)* | L1 *(-1.2)* |
|  | Reciprocal | Yes *(-1.0)* | Yes *(-5.9)* | Yes *(-1.1)* | No *(-1.0)* | Yes – |

Table 3: Hyperparameters of best performing models after quasi-random hyperparameter search and Bayesian optimization w.r.t. filtered MRR (on validation data). For each hyperparameter, we also give the reduction in filtered MRR for the best configuration that does not use this value of the hyperparameter (in parenthesis).

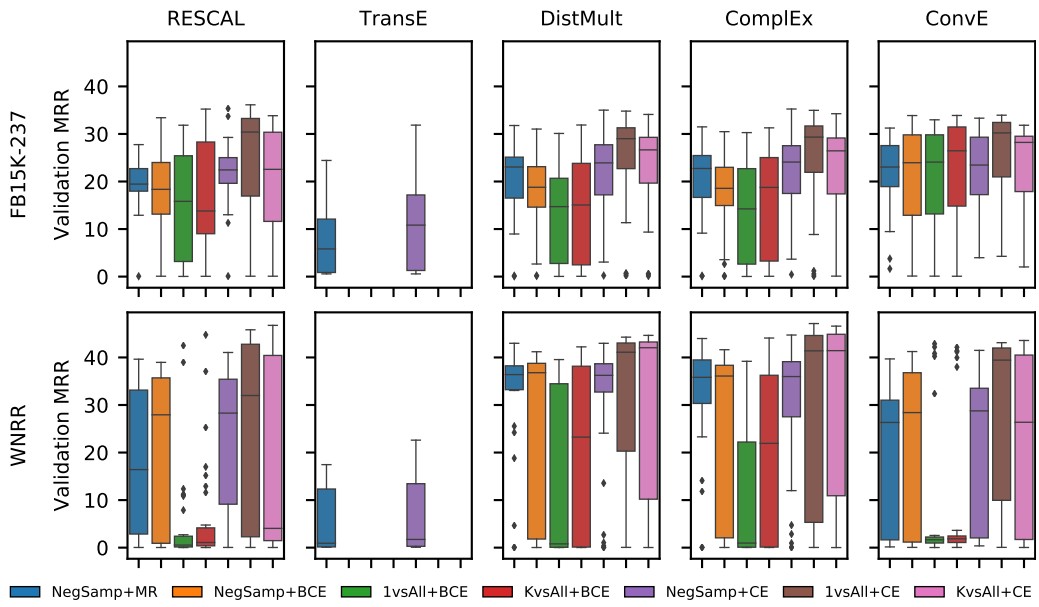

Figure 2: Distribution of filtered MRR (%) on validation data over the quasi-random hyperparameter configurations for different training type and loss functions.

we obtained when enforcing reciprocal relations. We can see that most hyperparameters appear to have moderate impact on model performance: many hyperparameter settings did not produce a significant performance improvement that others did not also produce. A clear exception here is the use of the loss function, which is consistent across models and datasets (always CE), partly with large performance improvements over alternative loss functions.

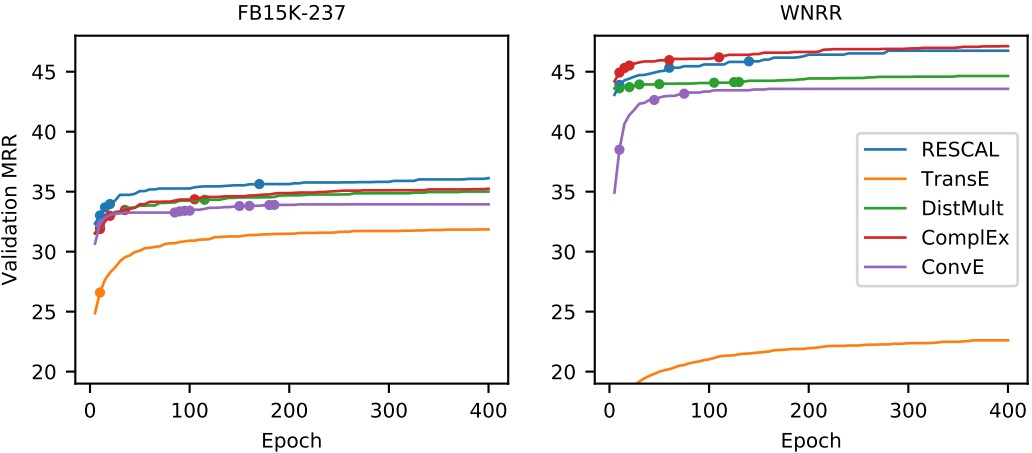

Figure 3: Best filtered MRR (%) on validation data achieved during quasi-random search as a function of the number of training epochs. Changes in the corresponding "best-so-far" hyperparameter configuration are marked with a dot.

**Impact of loss function.** In order to further explore the impact of the loss function, we show the distribution of the filtered MRR on validation data for every (valid) combination of training type and loss function in Figure 10. We found that the use of CE (the three rightmost bars) generally lead to better configurations than using other losses. This is surprising (and somewhat unsatisfying) in that we use the CE loss, which is a multi-class loss, for a multi-label task. Similar observations have been made for computer vision tasks (Joulin et al., 2016; Mahajan et al., 2018) though. Note that the combination of KvsAll with CE (on the very right in figure) has not been explored previously.

## 3.4   MODEL SELECTION

We briefly summarize the behavior of various models during model selection. Figure 3 shows the best validation MRR (over all quasi-random hyperparameter configurations) achieved by each model when trained for the specified number of epochs. The corresponding hyperparameter configuration may change as models were trained longer; in the figure, we marked such changes with a dot. We found that a suitable hyperparameter configuration as well as a good model can be found using significantly less than the 400 epochs used in our study. Nevertheless, longer training did help in many settings and some models may benefit when trained for even more than 400 epochs.

## 4   CONCLUSION

We reported the results of an extensive experimental study with popular KGE model architectures and training strategies across a wide range of hyperparameter settings. We found that when trained appropriately, the relative performance differences between various model architectures often shrunk and sometimes even reversed when compared to prior results. This suggests that (at least currently) training strategies have a significant impact on model performance and may account for a substantial fraction of the progress made in recent years. To enable an insightful model comparison, we recommend that new studies compare models in a common experimental setup using a sufficiently large hyperparameter search space. Our study shows that such an approach is possible with reasonable computational cost, and we provide our implementation as part of the LibKGE framework to facilitate such model comparison. Moreover, we feel that many of the more advanced architectures and techniques proposed in the literature need to be revisited to reassess their individual benefits.

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

## APPENDIX

**Evaluation metrics.** We formally define the evaluation metrics used in our study. Given a triple $(i, k, j)$, denote by $\mathrm{rank}(j|i, k)$ the filtered rank of object $j$, i.e., the rank of model score $s(i, k, j)$ among the collection of all pseudo-negative object scores

$$\{ s(i, k, j') : j' \in \mathcal{E} \text{ and } (i, k, j') \text{ does not occur in training, validation, or test data} \} .$$

If there are ties, we take the mean rank of all triples with score $s(i, k, j)$. Define $\mathrm{rank}(i|k, j)$ likewise. Denote by $\mathcal{K}^{\text{test}}$ the set of test triples. Then

$$\mathrm{MRR} = \frac{1}{2|\mathcal{K}^{\text{test}}|} \sum_{(i,k,j) \in \mathcal{K}^{\text{test}}} \left( \frac{1}{\mathrm{rank}(i|k, j)} + \frac{1}{\mathrm{rank}(j|i, k)} \right),$$

$$\mathrm{Hits@}K = \frac{1}{2|\mathcal{K}^{\text{test}}|} \sum_{(i,k,j) \in \mathcal{K}^{\text{test}}} \left( \mathbb{1}(\mathrm{rank}(i|k, j) \leq K) + \mathbb{1}(\mathrm{rank}(j|i, k) \leq K) \right),$$

where indicator $\mathbb{1}(E)$ is 1 if condition $E$ is true, else 0.

| Dataset | Entities | Relations | Train | Validation | Test |
|---------|----------|-----------|-------|------------|------|
| FB-237 | 14 505 | 237 | 272 115 | 17 535 | 20 466 |
| WNRR | 40 559 | 11 | 86 835 | 3 034 | 3 134 |

Table 4: Dataset statistics

| Hyperparameter | Quasi-random search | Bayesian optimization |
|----------------|---------------------|----------------------|
| Embedding size | $\{128, 256, 512\}$ | Value from Tab. 6 |
| Training type | $\{$NegSamp, 1vsAll, KvsAll$\}$ | Value from Tab. 6 |
|    Reciprocal | $\{$True, False$\}$ | Value from Tab. 6 |
|    No. subject samples (NegSamp) | $[1, 1000]$, log scale | $[1, 1000]$, log scale |
|    No. object samples (NegSamp) | $[1, 1000]$, log scale | $[1, 1000]$, log scale |
|    Label smoothing (KvsAll) | $[0.0, 0.3]$ | $[0.0, 0.3]$ |
| Loss | $\{$BCE, MR, CE$\}$ | Value from Tab. 6 |
|    Margin (MR) | $[0, 10]$ | $[0, 10]$ |
|    $L_p$-norm (TransE) | $\{1, 2\}$ | Value from Tab. 6 |
| Optimizer | $\{$Adam, Adagrad$\}$ | Value from Tab. 6 |
|    Batch size | $\{128, 256, 512, 1024\}$ | Value from Tab. 6 |
|    Learning rate | $[10^{-4}, 1]$, log scale | $[10^{-4}, 1]$, log scale |
|    LR scheduler patience | $[0, 10]$ | $[0, 10]$ |
| $L_p$ regularization | $\{$L1, L2, L3, None$\}$ | Value from Tab. 6 |
|    Entity emb. weight | $[10^{-20}, 10^{-5}]$ | $[10^{-20}, 10^{-5}]$ |
|    Relation emb. weight | $[10^{-20}, 10^{-5}]$ | $[10^{-20}, 10^{-5}]$ |
|    Frequency weighting | $\{$True, False$\}$ | Value from Tab. 6 |
| Embedding normalization (TransE) | | |
|    Entity | $\{$True, False$\}$ | Value from Tab. 6 |
|    Relation | $\{$True, False$\}$ | Value from Tab. 6 |
| Dropout | | |
|    Entity embedding | $[0.0, 0.5]$ | $[0.0, 0.5]$ |
|    Relation embedding | $[0.0, 0.5]$ | $[0.0, 0.5]$ |
|    Feature map (ConvE) | $[0.0, 0.5]$ | $[0.0, 0.5]$ |
|    Projection (ConvE) | $[0.0, 0.5]$ | $[0.0, 0.5]$ |
| Embedding initialization | $\{$Normal, Unif, XvNorm, XvUnif$\}$ | Value from Tab. 6 |
|    Std. deviation (Normal) | $[10^{-5}, 1.0]$ | $[10^{-5}, 1.0]$ |
|    Interval (Unif) | $[-1.0, 1.0]$ | $[-1.0, 1.0]$ |
|    Gain (XvNorm) | $1.0$ | $1.0$ |
|    Gain (XvUnif) | $1.0$ | $1.0$ |

Table 5: Hyperparameter search space used in our study. Settings that apply only to certain configurations are indicated in parenthesis.

| | RESCAL | TransE | DistMult | ComplEx | ConvE |
|---|---|---|---|---|---|
| Mean MRR | 36.1 | 31.5 | 35.0 | 35.3 | 34.3 |
| Embedding size | 128 (-0.5) | 512 (-3.7) | 256 (-0.2) | 256 (-0.3) | 256 (-0.4) |
| Training type | 1vsAll (-0.8) | NegSamp – | NegSamp (-0.2) | NegSamp (-0.3) | 1vsAll (-0.4) |
| Reciprocal | No (-0.5) | Yes (-9.5) | Yes (-0.3) | Yes (-0.3) | Yes – |
| No. subject samples (NegSamp) | – | 2 | 557 | 557 | – |
| No. object samples (NegSamp) | – | 56 | 367 | 367 | – |
| Label Smoothing (KvsAll) | – | – | – | – | – |
| Loss | CE (-0.9) | CE (-7.1) | CE (-3.1) | CE (-3.8) | CE (-0.4) |
| Margin (MR) | – | – | – | – | – |
| $L_p$-norm (TransE) | – | L2 | – | – | – |
| Optimizer | Adam (-0.5) | Adagrad (-3.7) | Adagrad (-0.2) | Adagrad (-0.5) | Adagrad (-1.5) |
| Batch size | 512 (-0.5) | 128 (-7.1) | 1024 (-0.2) | 1024 (-0.3) | 1024 (-0.4) |
| Learning Rate | 0.00063 | 0.04122 | 0.14118 | 0.14118 | 0.00373 |
| Scheduler patience | 1 (-0.5) | 6 (-3.7) | 9 (-0.2) | 9 (-0.3) | 5 (-0.4) |
| $L_p$ regularization | None (-0.5) | L2 (-3.7) | L3 (-0.2) | L3 (-0.3) | L3 (-0.4) |
| Entity emb. weight | – | $1.32^{-07}$ | $1.55^{-10}$ | $1.55^{-10}$ | $1.55^{-11}$ |
| Relation emb. weight | – | $3.72^{-18}$ | $3.93^{-15}$ | $3.93^{-15}$ | $7.91^{-12}$ |
| Frequency weighting | Yes (-0.5) | No (-11.0) | Yes (-0.3) | Yes (-0.3) | Yes (-1.5) |
| Embedding normalization (TransE) | | | | | |
| Entity | – | No | – | – | – |
| Relation | – | No | – | – | – |
| Dropout | | | | | |
| Entity embedding | 0.37 | 0.00 | 0.46 | 0.46 | 0.00 |
| Relation embedding | 0.28 | 0.00 | 0.36 | 0.36 | 0.10 |
| Projection (ConvE) | – | – | – | – | 0.19 |
| Feature map (ConvE) | – | – | – | – | 0.49 |
| Embedding initialization | Normal (-0.8) | XvNorm (-3.7) | Unif. (-0.2) | Unif. (-0.5) | XvNorm (-0.4) |
| Std. deviation (Normal) | 0.80620 | – | – | – | – |
| Interval (Unif) | – | – | [-0.85, 0.85] | [-0.85, 0.85] | – |

*FB15K-237*

Table 6: Hyperparameters of best performing models found with quasi-random hyperparameter optimization on FB15K-237. Settings that apply only to certain configurations are indicated in parenthesis. For each hyperparameter, we also give the reduction in filtered MRR for the best configuration that does not use this value of the hyperparameter (in parenthesis).

| | RESCAL | TransE | DistMult | ComplEx | ConvE |
|---|---|---|---|---|---|
| Mean MRR | 46.8 | 22.6 | 45.4 | 47.6 | 44.3 |
| Embedding size | 128 (-1.0) | 512 (-5.1) | 512 (-1.1) | 128 (-1.0) | 512 (-1.2) |
| Training type | KvsAll (-1.0) | NegSamp – | KvsAll (-1.1) | 1vsAll (-1.0) | KvsAll (-1.2) |
| Reciprocal | Yes (-1.0) | Yes (-5.9) | Yes (-1.1) | No (-1.0) | Yes – |
| No. subject samples (NegSamp) | – | 2 | – | – | – |
| No. object samples (NegSamp) | – | 56 | – | – | – |
| Label Smoothing (KvsAll) | 0.30 | – | 0.21 | – | -0.29 |
| Loss | CE (-2.0) | CE (-5.1) | CE (-2.4) | CE (-3.5) | CE (-1.4) |
| Margin (MR) | – | – | – | – | – |
| $L_p$-norm (TransE) | – | L2 | – | – | – |
| Optimizer | Adam (-1.2) | Adagrad (-5.8) | Adagrad (-1.5) | Adagrad (-1.5) | Adam (-1.4) |
| Batch size | 128 (-1.0) | 128 (-5.1) | 1024 (-1.1) | 512 (-1.0) | 1024 (-1.3) |
| Learning Rate | 0.00160 | 0.04122 | 0.25575 | 0.50338 | 0.00160 |
| Scheduler patience | 8 (-1.0) | 6 (-5.1) | 6 (-1.1) | 7 (-1.0) | 1 (-1.2) |
| $L_p$ regularization | L3 (-1.2) | L2 (-5.1) | L3 (-1.1) | L2 (-1.0) | L1 (-1.2) |
| Entity emb. weight | $3.82^{-20}$ | $1.32^{-07}$ | $1.34^{-10}$ | $1.48^{-18}$ | $3.70^{-12}$ |
| Relation emb. weight | $7.47^{-05}$ | $3.72^{-18}$ | $6.38^{-16}$ | $1.44^{-18}$ | $6.56^{-10}$ |
| Frequency weighting | No (-1.0) | No (-7.4) | Yes (-1.1) | No (-1.0) | No (-1.4) |
| Embedding normalization (TransE) | | | | | |
| Entity | – | No | – | – | – |
| Relation | – | No | – | – | – |
| Dropout | | | | | |
| Entity embedding | 0.00 | 0.00 | 0.12 | 0.05 | 0.00 |
| Relation embedding | 0.00 | 0.00 | 0.36 | 0.44 | 0.23 |
| Projection (ConvE) | – | – | – | – | 0.09 |
| Feature map (ConvE) | – | – | – | – | 0.42 |
| Embedding initialization | Unif. (-1.0) | XvNorm (-5.1) | Unif. (-1.3) | Unif. (-1.5) | XvNorm (-1.4) |
| Std. deviation (Normal) | – | – | – | – | – |
| Interval (Unif) | [-0.31, 0.31] | – | [-0.81, 0.81] | [-0.31, 0.31] | – |

_WNRR_

Table 7: Hyperparameters of best performing models found with quasi-random hyperparameter optimization on WNRR. Settings that apply only to certain configurations are indicated in parenthesis. For each hyperparameter, we also give the reduction in filtered MRR for the best configuration that does not use this value of the hyperparameter (in parenthesis).

| | | RESCAL | TransE | DistMult | ComplEx | ConvE |
|---|---|---|---|---|---|---|
| *FB15K-237* | Learning rate | 0.00074 | 0.00030 | 0.15954 | 0.18255 | 0.00428 |
| | Scheduler patience | 1 | 5 | 6 | 7 | 9 |
| | Entity reg. weight | – | – | $1.41^{-09}$ | $6.70^{-09}$ | $4.30^{-15}$ |
| | Relation reg. weight | – | – | $4.10^{-15}$ | $4.13^{-14}$ | $1.12^{-14}$ |
| | Entity emb. dropout | 0.43 | 0.00 | 0.42 | 0.50 | 0.00 |
| | Relation emb. dropout | 0.16 | 0.00 | 0.41 | 0.23 | 0.12 |
| | Projection dropout (ConvE) | – | – | – | – | 0.50 |
| | Feature map dropout (ConvE) | – | – | – | – | 0.49 |
| *WNRR* | Learning rate | 0.00085 | 0.25327 | 0.33127 | 0.52558 | 0.00162 |
| | Scheduler patience | 8 | 5 | 7 | 5 | 1 |
| | Entity reg. weight | $1.37^{-14}$ | $1.06^{-07}$ | $1.25^{-12}$ | $4.52^{-06}$ | $1.08^{-08}$ |
| | Relation reg. weight | $2.57^{-15}$ | $4.50^{-13}$ | $1.53^{-14}$ | $4.19^{-10}$ | $9.52^{-11}$ |
| | Entity emb. dropout | 0.00 | 0.25 | 0.37 | 0.36 | 0.00 |
| | Relation emb. dropout | 0.24 | 0.00 | 0.50 | 0.31 | 0.23 |
| | Projection dropout (ConvE) | – | – | – | – | 0.15 |
| | Feature map dropout (ConvE) | – | – | – | – | 0.33 |

Table 8: Hyperparameters of best performing models found after quasi-random hyperparameter optimization and Bayesian optimization. All other hyperparameters are the same as in Tables 6 (FB15K-237) and 7 (WNRR).

| | | RESCAL | TransE | DistMult | ComplEx | ConvE |
|---|---|---|---|---|---|---|
| *FB15K-237* | MRR | 36.1±0.3 | 31.5±0.1 | 35.0±0.0 | 35.3±0.1 | 34.3±0.1 |
| | Hits@10 | 54.3±0.5 | 49.8±0.2 | 53.5±0.1 | 53.9±0.2 | 52.4±0.2 |
| *WNRR* | MRR | 46.8±0.2 | 22.6±0.0 | 45.4±0.1 | 47.6±0.1 | 44.3±0.1 |
| | Hits@10 | 51.8±0.2 | 51.5±0.1 | 52.4±0.3 | 54.1±0.4 | 50.4±0.2 |

Table 9: Mean and standard deviation of the validation data performance of each model's best hyperparameter configuration when trained from scratch five times.

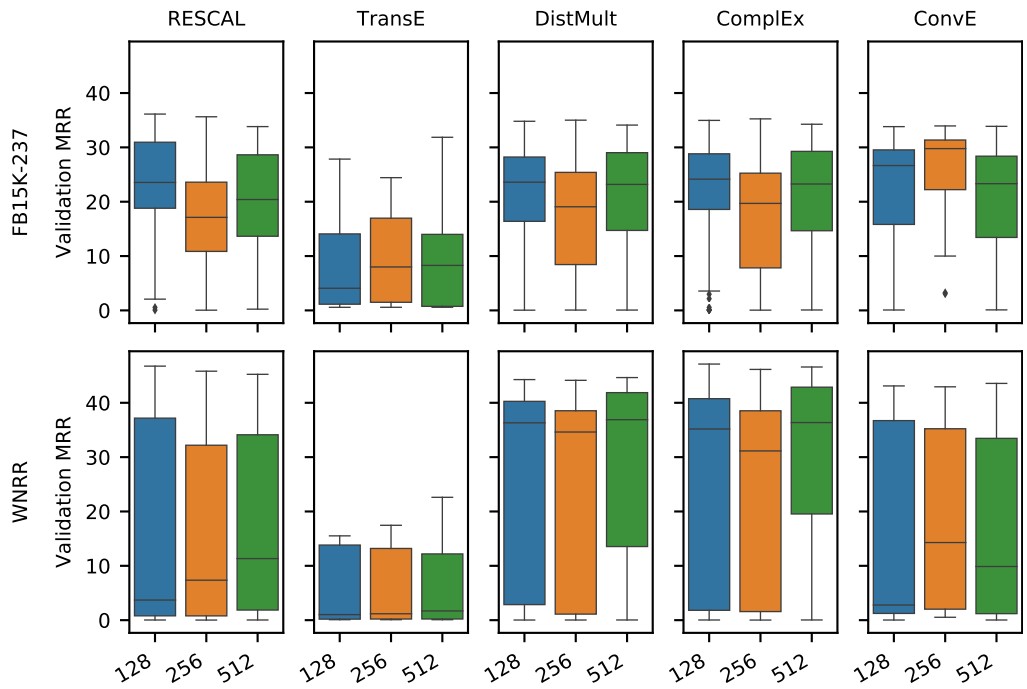

Figure 4: Distribution of filtered MRR (on validation data, quasi-random search only) for different embedding sizes (top row: FB15K-237, bottom row: WNRR)

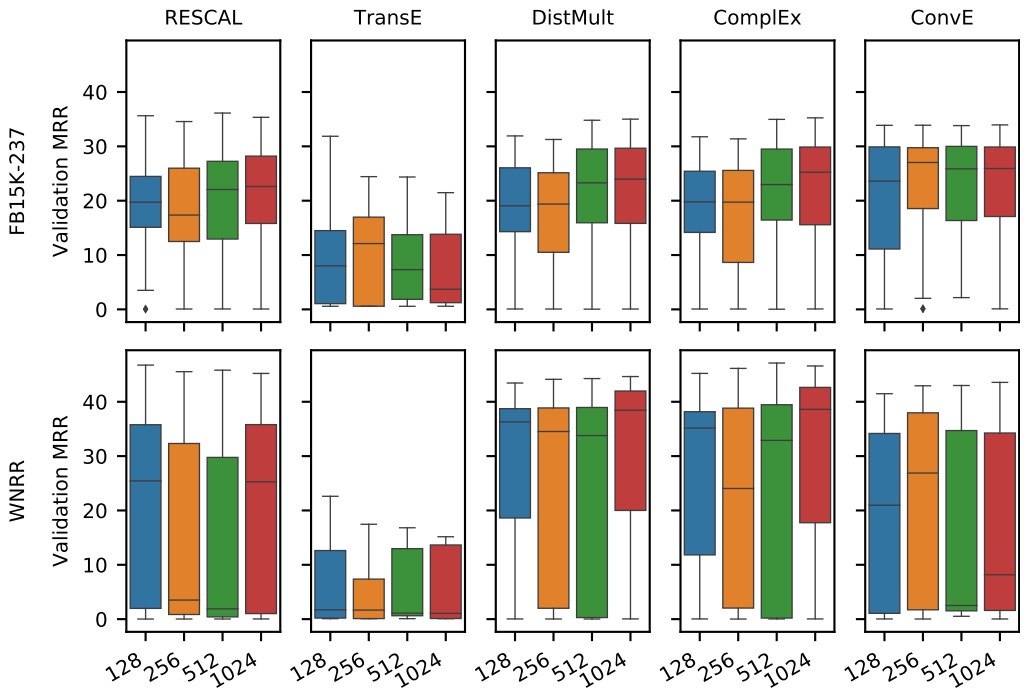

Figure 5: Distribution of filtered MRR (on validation data, quasi-random search only) for different batch sizes (top row: FB15K-237, bottom row: WNRR)

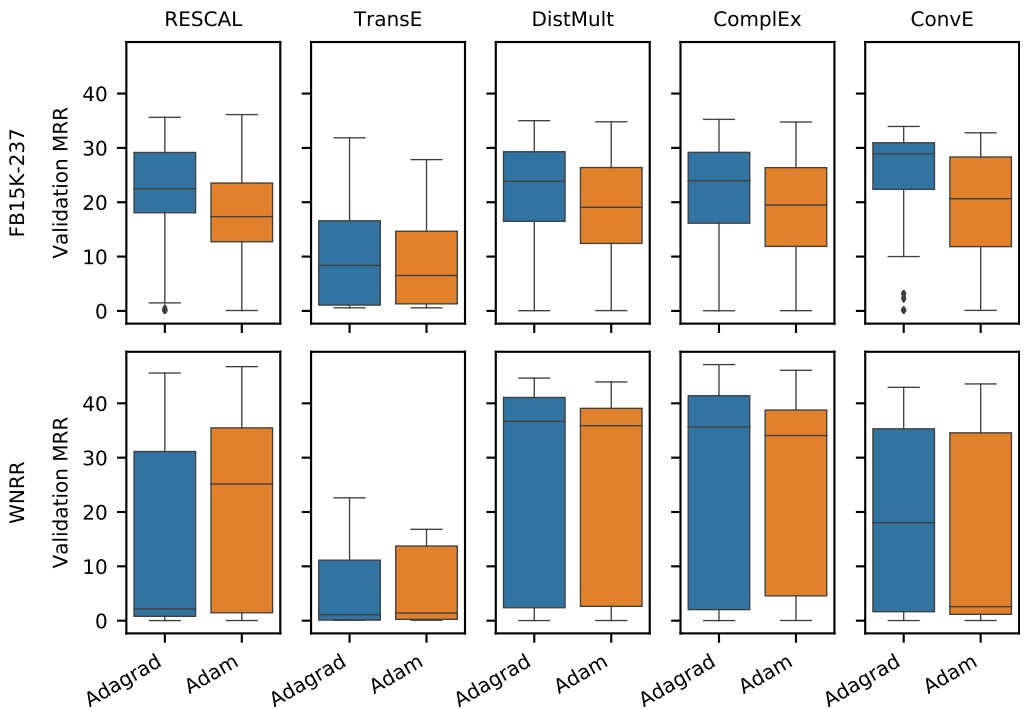

Figure 6: Distribution of filtered MRR (on validation data, quasi-random search only) for different optimizers (top row: FB15K-237, bottom row: WNRR)

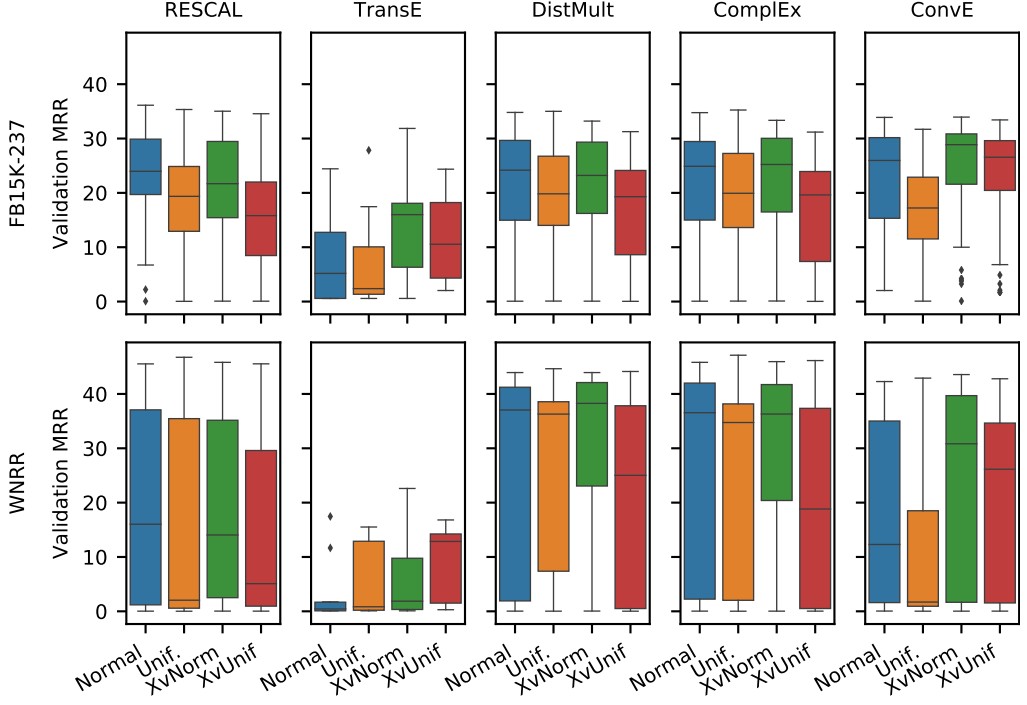

Figure 7: Distribution of filtered MRR (on validation data, quasi-random search only) for different initializers (top row: FB15K-237, bottom row: WNRR)

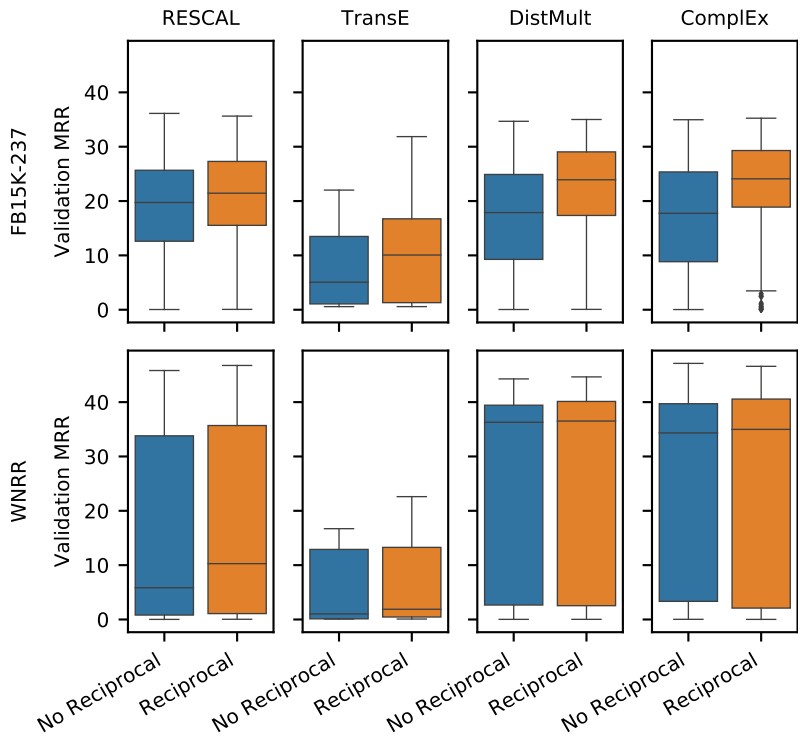

Figure 8: Distribution of filtered MRR (on validation data, quasi-random search only) with and without reciprocal relations (top row: FB15K-237, bottom row: WNRR)

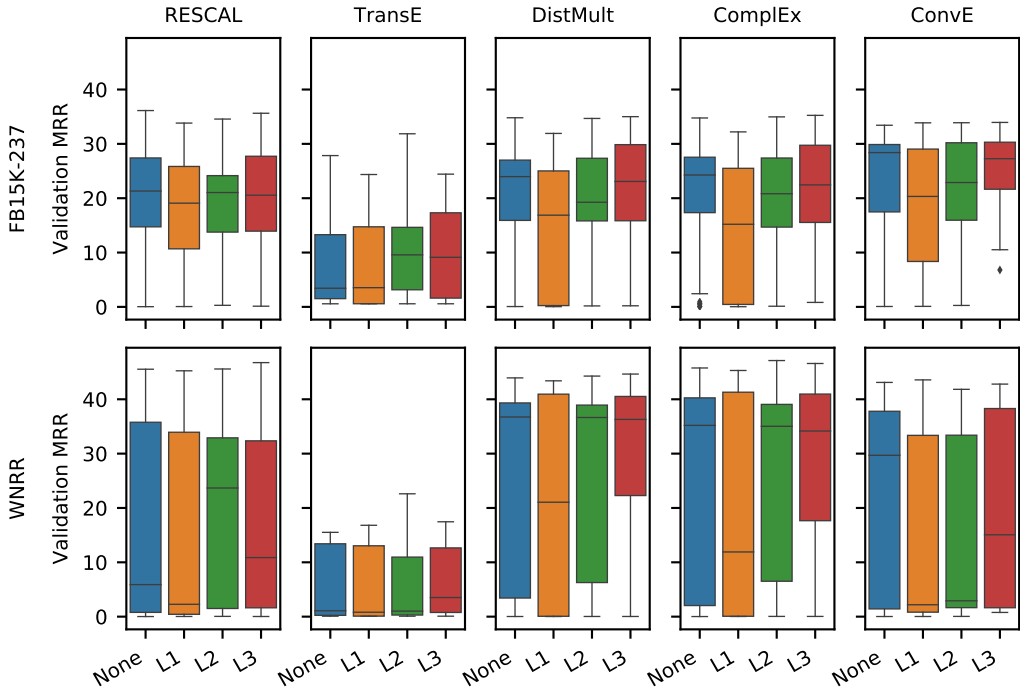

Figure 9: Distribution of filtered MRR (on validation data, quasi-random search only) for different penalty terms in the loss function (top row: FB15K-237, bottom row: WNRR)

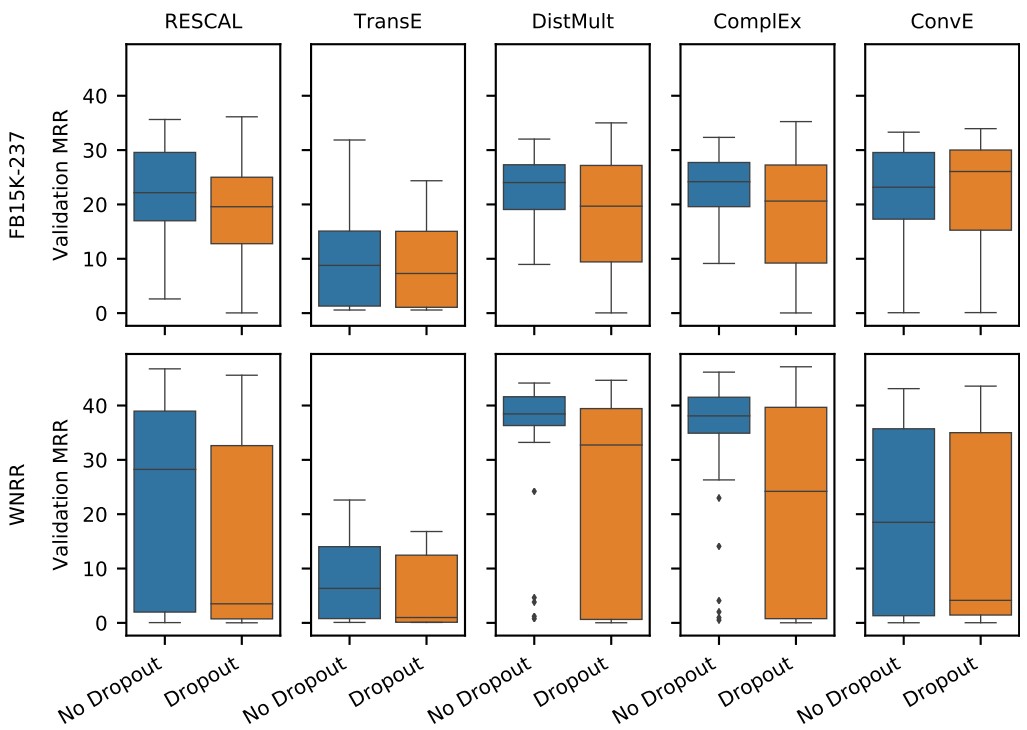

Figure 10: Distribution of filtered MRR (on validation data, quasi-random search only) with and without dropout (top row: FB15K-237, bottom row: WNRR)

