# OpenReview forum: "You CAN Teach an Old Dog New Tricks! On Training Knowledge Graph Embeddings"
_ICLR.cc/2020/Conference — Accept (Poster)_

### Official Review · AnonReviewer3 · 2019-10-24
**Official Blind Review #3**

**Rating:** 6

**Review:**

The paper presents an experimental study about some KGE methods. It argues that papers often propose changes in several different dimensions, such as model, loss, training, regularizer, etc., at once without providing a sufficient investigation about the individual components' contributions. The experimental study considers two datasets (FB15k-237 and WNRR) and five different models (RESCAL, TransE, DistMult, ComplEx, ConvE). The models were selected using a quasi-random hyperparameter search, followed by a short Bayesian optimization phase to fine-tune the parameters. The performance of the best models found during this hyperparameter search are compared to first published results for the same model, as well as to a small selection of recent papers. To analyse the influence of single hyperparameters, the best found configuration is compared to the best configuration which does not use this specific value for the given hyperparameter.

Overall, the paper adresses an important problem, as papers about new KGE methods often lack a clear separation of the individual changes' contribution. The experimental results show that older, simpler can compete with recently proposed models when trained properly. The intra-model comparison lacks statistical rigorousity, yet hints a few directions to further explore.

The experiments are based on a quasi-random hyperparameter search. While it is necessary for efficient exploration of larger search spaces [1], and should be the standard methodology for hyperparameter search of a new method, the interpretability of the comparison of two runs suffers. For the comparison between the trained models and previously published results, the sample size might be sufficient to draw the conclusions. However, the intra-model comparison, e.g. in Figure 2, are now comparing subsets of the runs which only comprise approx. (200/6) runs. Furthermore, the influence of random initialization is not accounted for. Another place where this can be witnessed is Table 3. Here, for some ablations, e.g. TransE + Reciprocal, no reduction is given. If I understood it correctly, this is due to not having a configuration which uses TransE and reciprocal relations. Also for the other ablations, it is unclear how statistically significant the reduction is.


Further Comments:
1. Please add the best published results for a specific model-dataset combination to table 2.
2. Do the plots in Figure 1 include the runs which were stopped after 50 epochs due to insufficient MRR?
3. Could you elaborate on the combination of KvsAll and CE?
4. The combination of subject and object triple scores has for instance been used in SimplE [2].


[1] Bergstra, James, and Yoshua Bengio. "Random search for hyper-parameter optimization." Journal of Machine Learning Research 13.Feb (2012): 281-305.
[2] Kazemi, Seyed Mehran, and David Poole. "Simple embedding for link prediction in knowledge graphs." Advances in Neural Information Processing Systems. 2018.


**Experience Assessment:**

I have read many papers in this area.

**Review Assessment: Checking Correctness Of Derivations And Theory:**

N/A

**Review Assessment: Checking Correctness Of Experiments:**

I carefully checked the experiments.

**Review Assessment: Thoroughness In Paper Reading:**

I read the paper thoroughly.

---

> ### Author Response · Authors · 2019-11-13
> **Thank you for your feedback and support**
>
> We thank you for your feedback and appreciate your support. In what follows, we briefly comment on the points raised in your review.
>
> 1. "For the comparison between the trained models and previously published results, the sample size might be sufficient to draw the conclusions. However, the intra-model comparison, e.g. in Figure 2, are now comparing subsets of the runs which only comprise approx. (200/6) runs."
>
> RESPONSE: We agree. The sample size for the intra-model comparisons is 30 (sometimes more); estimates such as the median MRR will be subject to variance. We chose to show the full distribution of the scores, however,
>   which gives a better picture than point estimates and exposes variance (e.g., bottom right of Fig 2.: high for first 2 bars, low for 3rd bar).
>
> 2. "The influence of random initialization is not accounted for."
>
> RESPONSE: Indeed, and it should be accounted for. We trained all models of Tab. 2 five times; the largest observed standard deviation for MRR was no more than 0.002, i.e., very low. This high stability is in line with prior studies; e.g., Sun et al. (2019). We will revise the paper and report mean and std. dev. of 5 training runs instead of the result of a single run.
>
> 3. "For some ablations, e.g. TransE + Reciprocal, no reduction is given."
>
> RESPONSE: We added the use of reciprocal relations to TransE and are rerunning the corresponding experiments. We indeed found improvements in TransE's performance due to the use of reciprocal relations and will update the paper accordingly. ConvE, on the other hand, cannot be used without reciprocal relations.
>
> 4. "Also for the other ablations, it is unclear how statistically significant the reduction is."
>
> RESPONSE: Especially the smaller effects may indeed not be significant. Note, however, that the "winning configuration" is consistently compared to a larger number of "ablation configurations". For example, 1/3 of the configurations use CE (winning configuration), but 2/3rd do not (ablation configurations).
>
> 5. "Please add the best published results for a specific model-dataset combination to table 2."
>
> RESPONSE: The performance of DistMult and ComplEx given in the "Large" section of Tab. 2 are---to the best of our knowledge---the best reported numbers for these two models. For TransE, we added a reference to the best reported number to the limitations section. For RESCAL and ConvE, our study reports the best numbers, again to the best of our knowledge.
>
> 6. "Do the plots in Figure 1 include the runs which were stopped after 50 epochs due to insufficient MRR?"
>
> RESPONSE: Yes. We may include statistics about the number of such configurations in the appendix. Would you consider this helpful? Generally, this statistic may serve as an indication of how easy it is to find a decent hyperparameter configurations for each model and dataset.
>
> 7. "Could you elaborate on the combination of KvsAll and CE?"
>
> RESPONSE: In KvsAll, examples correspond to "questions" such as (i,k,?) and all its true answers. We compute the cross entropy between the model distribution and the empirical distribution over all answers. The model distribution is given by the softmax distribution over the model scores s(i,k,?). The empirical distribution is given by the labels for all entities (for j-th entity: 1 if (i,k,j) in training data, 0 otherwise) normalized to sum to one. I.e., the empirical distribution for $n$ true answers to (i,k,?) assigns probability $1/n$ to each of the true answers and zero to everything else. In 1VsAll, on the other hand, examples directly correspond to training triples, as in prior work. The model distribution again correspond to softmax scores, but the data distribution now has a single 1 (the entity present in the current training triple). This matches the notion used by Kadlec et al.
>
> 8. "The combination of subject and object triple scores has for instance been used in simple [2]."
>
> RESPONSE: Thanks, we added the reference to the "reciprocal relations" section.

---

### Official Review · AnonReviewer2 · 2019-10-24
**Official Blind Review #2**

**Rating:** 6

**Review:**

Summary
========
The paper conducts a thorough analysis of existing models for constructing knowledge graph embeddings. It focuses on attempting to remove confounding aspects of model features and training regime, in order to better assess the merits of KGE models. The paper describes the reimplementation of five different KGE models, re-trained with a common training framework which conducts hyperparameter exploration. The results show surprising insights, e.g., demonstrating that a system from 2011, despite being the earliest of the KGE models analyzed, demonstrates competitive results over a more recent (2017) published model.

Overall Comments
===============
The paper, and the described software release specifically, represent a solid contribution to the area of knowledge graph embeddings. I agree with the basic premise of this paper’s analysis: in order to accelerate research in a maturing field (like knowledge graphs), it is important to be able to properly compare with older systems, removing artifacts that are due to general improvements in training and optimization techniques, from modeling specific changes. The report of the strong results from the RESCAL system, along with others, drive the point through. Furthermore, the paper is well-written and easy to follow, and should become a good reference for future works on KGEs.

Detailed comments
===============
Below are some detailed comments about specific parts of the paper, in order of importance:

1. The paper mentions disregarding “monolithic” models in the current analysis, primarily due to the expensive training of these models. It may, however, be the case that the future state-of-the-art models will be larger and slower to train (and, perhaps, of the monolithic type). Are there any limitations to the proposed experimental framework that would prevent running monolithic/large models?

2. Regarding the item above, if one were to look at the training curves for the exploration of the current 5 KGE models, is it possible that verify winning hyperparameter configurations earlier than the full training is complete. In my experience, it is often the case that with fewer than 1/10th steps of full training (well before convergence), it is possible to compare model configurations (relatively). For example, “Population-base training” (https://arxiv.org/abs/1711.09846, https://arxiv.org/abs/1902.01894) is one framework where fewer training steps are used to quickly learn good hyperparameter configurations. I’m wondering whether the KGE hyperparameter exploration training curves display similar early trends. Could a shortened training procedure produce sufficient information for learning good parameters, and potentially deal with larger/slower models?  In addition: would adopting population-based training be applicable to the proposed framework?

3. In Section 3.2, “Limitations”, there is a surprising comment that performance can be improved with further hyperparameter tuning. It is not clear how the authors found the configurations that produced the improved results. It would be helpful to clarify why the hyperparameter exploration proposed in the paper did not discover these improved configurations. Were the improved configurations outside of the range of considered values? Or would the exploration require more points to find the improved configuration?

4. In Section 3.3 “Best configurations (quasi-random search)”, specifically Table 3, the paper presents an ablation of independent hyperparameters, over the best configuration for each of the 5 models. This is a very interesting section. One further suggestion, however, is whether the paper could include the performance of each of the models on the _average_ best configuration. Although the paper describes losses for switching individual parameters to their second best values, it is unlikely that the losses are cumulative. So, for example, if we can take the average/majority best value for each parameter (embedding size = 512, batch size = 1024, training type = 1vsall, loss = CE, etc.), and collect results for that configuration. I think it would be interesting to know the difference between a model trained on a “collectively known good” set of parameters vs. a model and task specific tuned set of parameters.

5. In Section 2, “Evaluation”, HITS@k is not formally defined. Unfortunately, I have encountered slight variants of this metrics (e.g: (1) given a SINGLE correct label, HITS@k is the average rate of the label being present in the top k scored results, or (2) given ALL possible correct labels, HITS@k is the percentage of correct labels present within the top k scored results, etc.). It would be nice to precisely describe HITS@k in this work.

6. Caption for Table 2 does not contain a description for the “Recent” super-column.

7. In Section 3.3 “Best configuration (quasi-random search)” Space missing at “... Tables 6 and 7(in …”, between 7 and (.


**Experience Assessment:**

I have published one or two papers in this area.

**Review Assessment: Checking Correctness Of Derivations And Theory:**

I carefully checked the derivations and theory.

**Review Assessment: Checking Correctness Of Experiments:**

I carefully checked the experiments.

**Review Assessment: Thoroughness In Paper Reading:**

I read the paper thoroughly.

---

> ### Author Response · Authors · 2019-11-13
> **Thank you for your feedback and support**
>
> We thank you for your feedback and appreciate your support. In what follows, we briefly comment on the points raised in your review.
>
> 1. There are no such limitations in our experimental framework that we are aware of.
>
> 2. Good point! Generally, it may indeed be possible to short-circuit hyperparameter search but that is beyond our current study. Our framework is extensible, however, so that there shouldn't be any principal limitations in adding other hyperparameter optimization methods (and we'd like to include more). What we can do for the present study is to include plots that show model performance (e.g., best validation MRR obtained over all hyperparameter configurations) as a function of the epochs each configuration has been trained. The new plots will give information about how fast models can find good configurations and compare different models along these lines. Would you consider this helpful?
>
> 3. Our goal was to have a fair, balanced comparison, but not to find perfect hyperparameters. For example, the ComplEx result mentioned in the "Limitations" section uses a configuration which is indeed within our search space but was not found during our hyperparameter search. Of course, the more effort we spend on  hyperparameter search, the better models we may find.
>
> 4. We consciously did not report performance on an "collectively good configuration". A key point that we are trying to make is that there is no such configuration: any configuration will be good for some models but bad for others. The same argument extends to small search grids.
>
> 5. Thanks for bringing this to our attention. We use (1) and will include a formal definition of the metrics in the appendix.
>
> 6. Thanks, added.
>
> 7. Thanks, fixed.

---

> > ### Author Response · Authors · 2019-11-14
> > **Regarding comment 2**
> >
> > We have added Fig. 9 to our paper along the lines discussed above. The figure suggests that decent (but often not very good) configurations can be found by simply training for less than 400 epochs.

---

### Official Review · AnonReviewer1 · 2019-10-25
**Official Blind Review #1**

**Rating:** 8

**Review:**

Authors did an extensive experimental study over neural link prediction architectures that was never done before, in such a systematic way, by other works in this space. Their findings suggest that some hyperparameters, such as the loss being used, can provide substantial improvements to some models, and can be the reason of the significant improvements in neural link prediction accuracy the community observed in recent months.

This is a really interesting paper, and can really shine some light on what was going on in neural link prediction over recent years. It also provides a great overview of the field -- in terms of architectures, loss functions, regularizers, sampling strategies, data augmentation strategies etc. -- that is really needed right now in the field.

One concern I have is that the hyperparameter tuning strategy is not really described -- authors just say something along the lines of "we use av.dev", but for those unfamiliar with this specific hyperparameter optimiser this does not provide much information (e.g. what is a Sobol sequence? I had to look it up).

**Experience Assessment:**

I have published in this field for several years.

**Review Assessment: Checking Correctness Of Derivations And Theory:**

I carefully checked the derivations and theory.

**Review Assessment: Checking Correctness Of Experiments:**

I carefully checked the experiments.

**Review Assessment: Thoroughness In Paper Reading:**

I read the paper at least twice and used my best judgement in assessing the paper.

---

> ### Author Response · Authors · 2019-11-13
> **Thank you for your feedback and support**
>
> We thank you for your feedback and appreciate your support. We added a short explanation on quasi-random search to the main paper. We also plan to provide more details in the framework documentation (which also supports other methods for hyperparameter optimization).

---

### Public Comment · ~Bahare_Fatemi1 · 2019-09-29
**Two Questions/Comments**

Interesting work and interesting results. I have two questions/comments:
1- I was wondering if it is possible for the authors to include a figure representing the distribution of filtered MRR for different regularization techniques (or add some discussion on their relative performance)?
2- The authors attribute the use of reciprocal relations to Dettmers et al. 2018. I believe Dettmers et al. 2018 identified the leakage of the previous datasets due to the existence of inverse relations; the use of reciprocal relations for learning better embeddings was proposed in [1] and [2]. Also regarding “On the downside, the use of reciprocal relations means that a model does not provide a single triple score s(i, k, j) anymore (generally, s_{sub}(i, k, j) \neq s_{obj}(i, k, j); the discrepancy has not been studied yet).”, it has been proposed in [1] to considering the final score (s(i, k, j)) to be the average of the two scores (s_{sub}(i, k, j) and s_{obj}(i, k, j)) and it has been shown that this results in better performance compared to considering the final score to be either one of the scores (see SimplE vs SimplE-ignr in Table 1).
[1] https://papers.nips.cc/paper/7682-simple-embedding-for-link-prediction-in-knowledge-graphs
[2] http://proceedings.mlr.press/v80/lacroix18a.html

---

> ### Public Comment · ~Tim_Dettmers2 · 2019-09-30
> **Timeline of the use of reciprocal relations.**
>
> My original work (Dettmers et al., 2018) did not use reciprocal relations. However, when a bug was identified in my codebase [1], I made use of reciprocal relations to allow for the continued use of 1-K predictions. I updated my paper with new, corrected results, and I did not update my paper with the precise definitions of using reciprocal relations. As such, I would attribute the first defined use of reciprocal relations to Kazemi & Poole (2018) and Lacroix et. al (2018) as mentioned by Bahare Fatemi.
>
> [1] https://github.com/TimDettmers/ConvE/issues/18

---

> > ### Author Response · Authors · 2019-10-01
> > **Thanks for the clarification!**
> >
> > Thanks Tim for clearing this up. Also thanks to both Tim and Bahare Fatemi for the hint regarding the  SimplE paper, we will use those references.

---

> ### Author Response · Authors · 2019-10-01
> **Regularization techniques and reciprocal relations**
>
> Thank you for the comments. We will try to include data and a discussion about the different regularization techniques in the appendix. Also, thanks for the pointers to [1,2] with respect to introducing reciprocal relations; we'll add the corresponding references.

---

### Public Comment · ~Tim_Dettmers2 · 2019-09-30
**An important contribution. Source code needed.**

In my personal research, I found that there was always high variability between different code-bases and approaches. This is mostly due to (1) a variety of methods (batch size, loss, normalization, regularization etc.), and (2) some bugs in the evaluation procedure. I was not able to replicate some publications, for example, Kadlec et al., 2017 which is frustrating since such results can derail progress in the field. This work aims at a fair comparison of different knowledge graph completion by doing careful hyperparameter searches. As such this work can serve as a solid foundation and reference for future research. This is a very important contribution since the normalization of results across models allows for more precise calibration of promising research directions which allow for faster progress in this field of research.

However, this work would be much more impactful if it would be coupled to a software framework in which new models can be developed. It would be of critical important that such a framework would be peer-reviewed to ensure that the evaluation procedure and sampling techniques are performed correctly. I am happy to peer review code if the authors are willing to provide such code.

While the work would be strengthened significantly with the addition of a peer-reviewed codebase. I highly recommend this work to be accepted. Even without a peer-reviewed codebase, this work allows researchers to validate their personal codebases against results in this work.

[1] Knowledge Base Completion: Baselines Strike Back: https://arxiv.org/abs/1705.10744

---

> ### Author Response · Authors · 2019-10-01
> **Our framework will be released as open source**
>
> Thanks and absolutely! The paper will be accompanied with an open-source software framework (on GitHub, currently private), which implements the different training techniques, models, and hyperparameter search. We have briefly mentioned this in the "Reproducibility" section in the paper, but will say more on the project homepage once the paper is deanonymized. We may be able to provide a dump of the codebase upfront, but we are not sure if we can do so truly anonymously.

---

### Public Comment · ~Chen_Cai1 · 2019-10-05
**Group perspective**

Hello,

Very interesting and solid work. I would like to provide a different perspective for KGE that might be helpful. In paper [1], I give a group-theoretic treatment of KGE and connect different models as modeling relations in KG as elements in different groups.

From this perspective, RESCAL is actually quite powerful since it corresponds to $GL(n, R)$. Other more recent methods correspond to other "smaller" groups. I would not be surprised that if optimization is done right, RESCAL can perform as well as other methods due to its very general form.

To me, the most interesting thing is to quantify the improvement of modeling non-communicative relations (son’s wife is not wife’s son) by going from the abelian group to the non-abelian group.  RotatE can essentially model any finite abelian group (proved in [1]), and for non-abelian group, I saw three recent work [2][3][4]. It would be interesting to see under your evaluation platform, how much gain can we get by modeling non-abelian groups.

[1] Group Representation Theory for Knowledge Graph Embedding https://arxiv.org/abs/1909.05100
[2] Quaternion Knowledge Graph Embedding https://arxiv.org/abs/1904.10281
[3] Relation Embedding with Dihedral Group in Knowledge Graph https://arxiv.org/abs/1906.00687
[4] A Group-Theoretic Framework for Knowledge Graph Embedding (ICLR this year) https://openreview.net/forum?id=r1e30AEKPr

---

> ### Author Response · Authors · 2019-10-09
> **More models and datasets in the future**
>
> Hi Chen, thanks for the comments. We plan on extending our study to include more models and datasets. In addition, we will release our framework as open source, and since adding new models is straightforward, we hope this will help to keep a growing list of comparable results for KGE models.

---

### Public Comment · ~Apoorv_Umang_Saxena1 · 2019-10-11
**Best reported results of TransE**

Much needed analysis!
I just want to add that in the Appendix of RotatE [1], they have done an ablation study on TransE where they have achieved 0.333 MRR for TransE on FB15k-237 dataset using adversarial negative sampling. I have been able to reproduce the same using the code they provided. I felt this might be relevant for your paper, since your paper reports the best of 0.303

Thanks

[1] RotatE: Knowledge Graph Embedding by Relational Rotation in Complex Space https://arxiv.org/abs/1902.10197

---

> ### Author Response · Authors · 2019-10-18
> **Thanks**
>
> Hi Apoorv, thanks for bringing this to our attention!

---

### Public Comment · ~Jae_Hee_Lee2 · 2019-10-14
**Combining Cross Entropy with KvsAll**

Hi, first of all I want to say that I totally support this line of work. Researchers working on KGE (and also on other topics as well!) should deal the baselines fairly and pay as much attention to them as they do to their own models.

One thing that I found not so clear in the paper is how the cross entropy loss (CE) is combined with KvsAll. (Note that this combination is used for the most of the best performing models on WN18RR in Table 3 of the paper).
It is clear to me that, based on the loss definition in [Kadlec et al., 2017], CE can be combined with 1vsAll. But it is not straight forward how CE can be combined with KvsAll, as claimed in line 7-8, page 4: "CE ...  has also been used in the multi-label setting (KvsAll)". Please either add a reference to the claim or give a more detailed explanation.

---

> ### Author Response · Authors · 2019-10-18
> **Cross entropy with KvsAll**
>
> Hi Jae, thanks for the support. As for your question, we are not using the CE definition of Kadlec et al., but directly the notion of cross entropy. Consider training point (i,k,j) and task (i,k,?) for which we compute the cross entropy between the model distribution, i.e. the softmax distribution of the scores s(i,k,?), and the data, i.e. the empirical distribution. For 1vsAll, the empirical distribution assigns probability 1 at position j, i.e., the true label, the rest is zero. This matches the notion used by Kadlec et al. For KvsAll, the empirical distribution for n true answers to (i,k,?) assigns probability 1/n to each of the true answers, zero to everything else. Does this clarify?

---

> > ### Public Comment · ~Jae_Hee_Lee2 · 2019-10-22
> > **Cross entropy with KvsAll**
> >
> > Yes, the question has been answered. Thanks.
> > (It is still puzzling though why  KvsAll & CE performs sometimes better than KvsAll & BCE, as they both are based on the local closed world assumption.)

---

### Decision · Program_Chairs · 2019-12-19

**Decision:**

Accept (Poster)

**Comment:**

The authors analyze knowledge graph embedding models for multi-relational link predictions. Three reviewers like the work and recommend acceptance. The paper further received several positive comments from the public. This is solid work and should be accepted.